# Proteasomal Degradation of Mutant Huntingtin Exon1 Regulates Autophagy

**DOI:** 10.3390/cells15010068

**Published:** 2025-12-30

**Authors:** Austin Folger, Chuan Chen, Phasin Gonzalez, Sophia L. Owutey, Yanchang Wang

**Affiliations:** 1Department of Biomedical Sciences, College of Medicine, Florida State University, 1115 West Call Street, Tallahassee, FL 32306-4300, USA; austin.folger@med.fsu.edu (A.F.); sophia.owutey@med.fsu.edu (S.L.O.); 2College of Biological Sciences, Hebei University, Baoding 071002, China; hyperchuan@aliyun.com; 3College of Medicine, Florida State University, 1115 West Call Street, Tallahassee, FL 32306-4300, USA; plg21b@med.fsu.edu

**Keywords:** misfolded proteins, mutant huntingtin exon1 (mHttEx1), proteasome, Cdc48 complex, IBophagy, autophagy

## Abstract

**Highlights:**

**What are the main findings?**
Yeast mutants defective in proteasomal degradation of soluble mutant huntingtin exon1 (mHttEx1) display impaired IBophagy, a selective autophagy pathway for misfolded protein inclusion bodies.Specific Cdc48 Ubx cofactors are required for efficient proteasomal degradation of mHttEx1 and IBophagy.Inhibition of proteasome function in yeast cells abolishes IBophagy.

**What is the implication of the main finding?**
The strong correlation between proteasomal degradation of soluble mHttEx1 and IBophagy indicates the negative role of mHttEx1 in autophagy.Our finding that IBophagy depends on the degradation of soluble mHttEx1 may represent an additional mechanism underlying mHttEx1-induced cytotoxicity.

**Abstract:**

Accumulation of misfolded proteins is implicated in neurodegenerative diseases. One of these is Huntington’s disease, which is caused by an expansion of trinucleotide (CAG) repeats in exon 1 of huntingtin gene (*HTT*). This expansion results in the production of mutant huntingtin exon1 protein (mHttEx1) containing polyglutamine tracks that is prone to cytotoxic aggregation. These mHttEx1 aggregates range from small soluble aggregates to large insoluble inclusion bodies. The mechanisms to clear mHttEx1 aggregates include ubiquitin-dependent proteasomal degradation and autophagy. For the proteasomal degradation of mHttEx1, ubiquitinated protein is first recognized by the Cdc48 complex for extraction and unfolding. For autophagy, mHttEx1 inclusion bodies are engulfed by an autophagosome, which fuses with the vacuole/lysosome and delivers cargo for vacuolar degradation. We name this autophagy IBophagy. In this study, we further show that the ubiquitination of mHttEx1 by the E3 ligase San1, its extraction and unfolding by the Cdc48 complex, and subsequent proteasomal degradation are all essential steps for mHttEx1 IBophagy in budding yeast, revealing a new layer of autophagy regulation and mHttEx1 cytotoxicity.

## 1. Introduction

Protein folding is a tightly regulated process, and multiple mechanisms ensure correct folding. However, protein misfolding still occurs spontaneously and can be aggravated by mutations, aging, environmental stresses, and translational errors [1]. Misfolded proteins are prone to aggregation, generating small soluble oligomers or large insoluble inclusion bodies (IBs) [2]. This aggregation is primarily driven by exposed hydrophobic surfaces [3,4]. Misfolded protein aggregates show cytotoxicity and are linked to numerous neurodegenerative disorders, such as Huntington’s and Alzheimer’s diseases [5,6]. Cellular mechanisms to combat this toxicity include the degradation of toxic proteins by ubiquitin-proteasome system (UPS) and macroautophagy [7,8,9]. Thus, it is pertinent to understand how these systems respond to the accumulation of misfolded proteins.

Huntington’s disease is a dominant neurodegenerative disorder caused by the expansion of glutamine-encoding trinucleotide (CAG) repeats in exon 1 of the huntingtin gene (*HTT*). Individuals with greater than 39 CAG repeats in this exon develop Huntington’s disease [10]. The expansion of CAG repeats in *HTT* results in production of misfolded Htt exon 1 protein with a polyQ track (mHttEx1) that is prone to aggregation [11]. The mHttEx1 fragment includes the 17 N-terminal amino acids of Htt, the expanded polyQ tract, and a proline-rich domain, which is a result of aberrant splicing [12]. Cytotoxicity arises when HttEx1 accumulates in the cell. mHttEx1 production in animal models results in symptoms of Huntington’s disease [13]. In addition, the onset of symptoms in aged humans with mHttEx1 is likely attributed to weakened protein quality control systems [14]. Thus, mHttEx1 has been an ideal substrate to study misfolded protein aggregation, clearance, and cytotoxicity.

Macroautophagy (hereafter referred to as autophagy) is an evolutionarily conserved pathway responsible for the degradation and recycling of cellular components [15]. This pathway involves the sequestration of cytoplasmic material into double-membraned autophagosomes, which subsequently fuse with the vacuole or lysosome, where the cargo is degraded and recycled by resident hydrolases [16]. Autophagy can function in either a selective or non-selective manner. Non-selective autophagy is typically induced under nutrient-deprived conditions and involves the bulk, random engulfment of cytosolic constituents for vacuolar or lysosomal degradation [17,18]. Selective autophagy, however, only targets specific proteins/structures for degradation through selective autophagy receptors (SARs) [19,20]. Selective autophagy has been implicated in the degradation of misfolded protein aggregates, including mHttEx1 [8,21]. We recently showed that IB autophagy (IBophagy) utilizes SARs to confer specificity for IBs formed by mHttEx1 in yeast cells [22]. To detect IBophagy, we induced mHttEx1 IB formation in galactose media, but IBophagy was observed only after adding glucose for the shutoff of mHttEx1 production. Therefore, an open question is why IBophagy only occurs after glucose addition. Interestingly, previous studies showed that mHttEx1 itself impairs autophagy [23,24,25], while it is unclear how mHttEx1 affects IBophagy.

In addition to autophagy, misfolded proteins can also be degraded by the UPS [26]. This system begins with the ubiquitination of target proteins. For misfolded proteins, the responsible E3 ubiquitin ligases in budding yeast include nuclear-localized San1 and cytoplasmic Ubr1 [27,28]. The efficient degradation of misfolded proteins also requires SUMOylation system and Cdc48 segregase [29,30]. Cdc48 ATPase assembles into a hexamer that translocates polypeptides through its central pore [31]. The yeast Cdc48 complex is composed of the Cdc48 ATPase together with its cofactors Npl4 and Ufd1 [32]. Npl4 recognizes ubiquitinated protein substrates, after which Cdc48 extracts ubiquitinated misfolded proteins from aggregates, unfolds them, and transfers them to the proteasome [31]. Regulators for the Cdc48 complex also include seven Ubx proteins in budding yeast, which contain a ubiquitin-interacting motif (UIM) and a Cdc48 binding motif (UBX domain) to bridge ubiquitinated substrates and Cdc48 [33,34]. We previously showed that proteasomal degradation of mHttEx1 depends on E3 ubiquitin ligase San1 as well as the Cdc48 complex in yeast cells [29]. However, it is unclear if the Cdc48 cofactors are all required for this degradation. Moreover, the role of the UPS pathway in IBophagy remains unknown.

In this work, we investigated the requirements of UPS components as well as Cdc48 and its cofactors in proteasomal degradation of mHttEx1 in yeast cells. We further analyze the role of this degradation in IBophagy. Our work utilizes a mHttEx1 protein with 103 polyQ (Htt103QP) as a model misfolded protein. We previously showed that E3 ligases San1, but not Ubr1, is required for proteasomal degradation of Htt103QP [29]. In this work, we found that San1, rather than Ubr1, is required for Htt103QP IBophagy. Our previous work also showed that the Cdc48 complex is required for proteasomal degradation of Htt103QP, and here we found that all three essential components of the Cdc48 complex, Cdc48, Npl4, and Udf1, are required for Htt103QP IBophagy. We further showed that three of seven Ubx proteins, the nonessential Cdc48 Ubx cofactors, are required for both IBophagy and proteasomal clearance of Htt103QP. In addition, inhibition of proteasomal activity by MG132 also prevents Htt103QP IBophagy. Therefore, our results established a strong correlation between proteasomal degradation and autophagic clearance of Htt103QP. Since Htt103QP presents in two pools: soluble Htt103QP and insoluble IBs, our results indicate that the proteasomal degradation of soluble Htt103QP facilitates IBophagy, likely by alleviating inhibition of autophagy by soluble Htt103QP.

## 2. Materials and Methods

### 2.1. Yeast Strains, Growth Conditions, and Plasmids

All yeast strains used in this research are in W303 background. Their relevant genotypes are listed in Appendix A. Standard tetrad dissection was used for yest strain construction. The *P_GAL_FLAG-Htt103QP-GFP* plasmid, originally obtained from the Lindquist lab, is a galactose-inducible construct encoding a FLAG- and GFP-tagged Htt103QP fragment [35]. After linearization, this plasmid was transformed into yeast cells for its insertion into the yeast genome. Yeast cells with the integration were selected using *URA3* marker and uracil dropout plates. Yeast peptone dextrose (YPD) was used for the growth of yeast cells. Yeast extract/peptone media (YEP) supplied with galactose was used to induce *FLAG-Htt103QP-GFP* expression and the subsequent inclusion body (IB) formation.

MG132 (Sigma-Aldrich, St. Louis, MO, USA), is a proteasome inhibitor, but it does not penetrate yeast cell wall and membrane. To enhance permeability of yeast cells, we added 0.1% L-proline and 0.003% SDS into the medium for 3 h prior to MG132 treatment [36]. Our previous studies confirmed that this protocol enables the inhibition of yeast proteasome function by 75 µM MG132 [37,38].

### 2.2. Detailed Protocol for IBophagy Induction

For the IBophagy assay, saturated yeast cells containing *pep4∆ VPH1-mApple P_GAL_-FLAG-Htt103QP-GFP* in YPD (yeast extract-peptone-dextrose) were diluted into galactose media (yeast extract, peptone, galactose) at 1:1000 and incubated for 16 h at 30 °C to induce Htt103QP IB formation. Glucose was subsequently added to a final concentration of 2% to repress *GAL1* promoter and halt Htt103QP production, thereby inducing IBophagy. To eliminate the potential effect of cell growth on autophagy, 200 mM hydroxyurea (HU) (Ambeed, Buffalo Grove, IL, USA) was also included to arrest yeast cell cycle in S-phase. Cells were harvested prior to glucose addition (time 0) and two hours after glucose treatment. Collected cells were washed with water, resuspended in 1× PBS, and immediately analyzed by fluorescence microscopy.

### 2.3. Fluorescence Imaging and Analysis

Htt103QP IB formation and autophagic clearance were analyzed using a fluorescence microscope (Keyence BZ-X700; Keyence of America, Itasca, IL, USA). Prepared cells were imaged with a 60× objective using appropriate filter sets for mApple and GFP. Z-stack images were acquired at 0.2-μm intervals and processed using BZ-X800 software (1.1.2.4.) to generate composites images. Autophagic activity was evaluated by assessing average GFP density inside vacuoles through automated hybrid cell counting with BZ-X800 software. Briefly, vacuolar regions were defined by extracting the mApple signal from the composite image, and the average integrated GFP intensity within mApple-labelled areas was measured in more than 50 cells per condition.

### 2.4. Western Blotting

Protein extracts were prepared using an alkaline extraction protocol and separated by 10% SDS–PAGE. Anti-FLAG antibody was obtained from Sigma-Aldrich (St. Louis, MO, USA); anti-Pgk1 antibody was purchased from Invitrogen (Waltham, MA, USA). Horseradish peroxidase–conjugated goat anti-mouse IgG secondary antibodies were from Cell Signaling Technology (Danvers, MA, USA). Following enhanced chemiluminescence (ECL; PerkinElmer, Waltham, MA, USA), membranes were imaged using a Bio-Rad ChemiDoc system (Hercules, CA, USA).

### 2.5. Statistical Analysis

Data are presented as mean ± standard error of the mean (SEM). For each strain, fluorescence average integrated density was calculated by quantifying vacuolar GFP intensity using BZ-X800 software. Statistical analyses were performed using GraphPad software (10.0.1). One-way or two-way ANOVAs was applied as appropriate to determine *p*-values, with the specific test indicated in the figure legends. Statistical significance was defined as *p* < 0.05 (*).

## 3. Results

### 3.1. Loss of E3 Ubiquitin Ligase San1 Blocks IBophagy of Htt103QP

As a model misfolded protein, Htt103QP forms IBs in yeast cells after overnight induction of *Htt103QP* expression in galactose media. Interestingly, IBophagy occurs only after glucose is added to shut off Htt103QP production [22]. We previously showed that San1 E3 ligase is essential for the ubiquitination and proteasomal degradation of Htt103QP [29]. Thus, we were curious whether San1 is required for Htt103QP IBophagy. To examine Htt103QP IBophagy, we utilized a previously constructed integrating plasmid *P_GAL_-FLAG-Htt103QP-GFP*, which contains a galactose-inducible promotor *GAL* and the first exon of *HTT* gene with 103 CAG expansion in fusion with *FLAG* and *GFP* gene. Yeast strains harboring this integrating plasmid produce FLAG-Htt103QP-GFP (hereafter Htt103QP) in the presence of galactose, forming Htt103QP IBs [37]. To analyze IBophagy, yeast cells used in this study also produce Vph1-mApple as a vacuolar marker, and are lacking the Pep4 protease (*pep4Δ*) to avoid protein degradation inside the vacuole [39,40].

To exclude the effect of cell cycle on autophagy, we added hydroxyurea (HU) to block cell cycle in our previous studies [22]. However, the HU-induced DNA replication stress may induce autophagy [41,42,43]. Therefore, we first examined the effect of HU on IBophagy in yeast cells. *pep4Δ VPH1-mApple* yeast strains with integrated *P_GAL_-FLAG-Htt103QP-GFP* plasmid were cultured in galactose-containing media for 16 h at 30 °C to induce IB formation. Then, HU alone or in combination with glucose was added to the medium. The appearance of GFP signal in the vacuole, which is marked by Vph1-mApple, was used as an indicator of successful IBophagy. The addition of HU alone did not cause any noticeable change in vacuolar GFP signal, but addition of both HU and glucose increased vacuolar GFP signal significantly (Figure 1A,B). The result suggests that treatment with HU alone does not induce IBophagy under our experimental conditions, likely because short-term HU exposure is insufficient to activate autophagy in yeast. We therefore applied this protocol to identify additional IBophagy regulators.

Because San1 E3 ubiquitin ligase is required for Htt103QP proteasomal degradation [29], we first tested whether this degradation is required Htt103QP IBophagy. For this purpose, wild-type (WT) and *san1Δ* cells were grown overnight in galactose media to induce Htt103QP IB formation. Glucose was then added to halt Htt103QP production and induce IBophagy. HU was also added to 200 mM to arrest the cells in S-phase. Two hours after glucose addition, most WT cells exhibited vacuolar localization of GFP, indicating successful Htt103QP IBophagy. However, minimal GFP vacuolar localization was observed in *san1Δ* cells (Figure 1C). Ubr1, the cytoplasmic E3 ligase for misfolded proteins, is dispensable for proteasomal degradation of Htt103QP [29]. Interestingly, IBophagy occurred normally in *ubr1Δ* cells as in WT cells. We measured the intensity of vacuolar GFP before and after glucose addition and found a significant increase in WT and *ubr1Δ* cells, indicating IBophagy, but no significant increase was observed in *san1Δ* cells (Figure 1D). Consistently, cells with IBs sharply declined in WT and *ubr1Δ* cells after glucose addition (Figure 1E), but we observed increased number of cells displaying vacuolar GFP signal (Figure 1F), consistent with successful IBophagy. In clear contrast, these changes were absent in *san1Δ* cells, where the number of cells with an IB or vacuolar GFP remains consistent. Collectively, these findings demonstrate that San1, but not Ubr1, is required for both proteasomal degradation of Htt103QP and IBophagy.

### 3.2. The Cdc48 Complex Is Required for IBophagy of Htt103QP

Our lab has shown that the Cdc48 complex promotes the proteasomal degradation of Htt103QP [29]. Thus, we tested if the Cdc48 complex is also required for Htt103QP IBophagy. Because the Cdc48 complex is essential in budding yeast, we used three temperature-sensitive mutants: *cdc48-3*, *npl4-1*, and *ufd1-2*. These mutations were introduced into *pep4Δ* cells producing Htt103QP and Vph1-mApple. We then grew cells in galactose media overnight at 25 °C to induce Htt103QP IB formation and shifted to 37 °C for one hour to inactivate the Cdc48 complex. Glucose and HU were then added to halt Htt103QP production and arrest the cell cycle while inducing IBophagy. We found that all mutants exhibited persistent presence of Htt103QP IBs and minimal GFP vacuolar localization, indicating defective IBophagy in these mutants (Figure 2A). Although WT cells show greater vacuolar GFP intensity and fewer IBs after glucose addition, the cells with defective Cdc48 complex showed no significant change in vacuolar GFP intensity and maintained a relatively consistent proportion of cells with either IBs or vacuolar GFP (Figure 2B–D). Therefore, the Cdc48 complex is required for both proteasomal degradation and IBophagy of Htt103QP.

### 3.3. The Proteasomal Degradation and IBophagy of Htt103QP Depends on Specific Ubx Proteins

In budding yeast, the Cdc48 complex uses seven Ubx domain-containing cofactors to facilitate the recruitment of ubiquitinated proteins [44,45]. Thus, some Ubx proteins are likely required for Htt103QP proteasomal degradation and IBophagy. To test this, we examined their requirements in Htt103QP IBophagy. For this purpose, we deleted *UBX1*–*7* genes in yeast cells with *pep4Δ VPH1-mApple P_GAL_-FLAG-Htt103QP-GFP*. Using the described IBophagy protocol, we found that vacuolar Htt103QP-GFP localization was deficient in *ubx1Δ*, *ubx3Δ*, *ubx4Δ*. A slight increase in vacuolar GFP was observed in *ubx7Δ* cells after IBophagy induction, while the increase in vacuolar GFP signal in *ubx2Δ*, *ubx5Δ*, and *ubx6Δ* mutant cells was similar to WT cells (Figure 3A,B). In addition, *ubx1Δ*, *ubx3Δ*, *ubx4Δ* mutants displayed a relatively consistent proportion of cells with either IBs or vacuolar GFP after glucose addition (Figure 3C,D). These results indicate that specific Ubx cofactors, Ubx1, Ubx3, and Ubx4, are required for the efficient Htt103QP IBophagy.

Because the Ubx protein family bridges ubiquitinated proteins to the Cdc48 complex for extraction/unfolding and eventual proteasomal degradation, we investigated their requirement for Htt103QP proteasomal degradation. The strains used for this experiment contain *pep4Δ,* which reduces autophagic degradation of Htt103QP. Our previous studies using expression shut-off assay show proteasome-dependent Htt103QP degradation, since proteasome inhibitor MG132 blocks this degradation [29]. We applied the same protocol to analyze the role of Ubx proteins in proteasomal degradation of Htt103QP. For this experiment, we grew WT and *ubxΔ* mutant cells in raffinose media to log phase before inducing *HTT103QP* expression with galactose for one hour. Expression was shut off by adding glucose, and we then examined Htt103QP degradation kinetics over time. As shown previously, Htt103QP was readily degraded in WT cells over 180 min [29], but delayed Htt103QP protein degradation was observed in *ubx1Δ*, *ubx3Δ*, and *ubx4Δ* cells. In contrast, the delay was not obvious in *ubx2Δ*, *ubx5Δ*, *ubx6Δ,* and *ubx7Δ* cells, indicating only certain Ubx proteins are required for proteasomal degradation of Htt103QP (Figure 4A,B). Strikingly, all three *ubxΔ* mutants exhibiting impaired proteasomal Htt103QP degradation, *ubx1Δ*, *ubx3Δ*, and *ubx4Δ,* also showed defective IBophagy of Htt103QP (Figure 3). This strong correlation suggests that proteasomal degradation of Htt103QP or some other protein is required for IBophagy.

### 3.4. Proteasome Function Is Required for Htt103QP IBophagy

Our results so far reveal the strong correlation between proteasomal Htt103QP degradation and IBophagy. Thus, we examined whether proteasome activity is required for IBophagy. To this end, we grew cells overnight in galactose media to induce IB formation. After treatment of the cells with L-proline and SDS to increase cell permeability, MG132 was added to 75 μM for 30 min to inhibit proteasome activity [36,37]. Then, glucose was added to stop Htt103QP production and induce IBophagy. We found that the vacuolar localization of Htt103QP-GFP as well as the disappearance of Htt103QP IBs was blocked in cells treated with MG132 (Figure 5A–D), indicating defective IBophagy. Previous studies showed that yeast cells treated with MG132 triggers selective autophagy of proteasomes [46], indicating that inhibition of proteasome function per se does not affect autophagy. Therefore, this result indicates that proteasomal degradation of Htt103QP or some other protein is likely required for IBophagy.

## 4. Discussion

UPS and Cdc48 segregase contribute to the clearance of soluble mHttEx1 [9,29], but constant expression of mHttEx1 or other misfolded proteins results in the formation of large insoluble IBs. We previously demonstrated autophagic clearance of mHttEx1 IBs through IBophagy [22]. However, IBophagy began only when Htt103QP production was shut off with glucose. To determine the underlying mechanism, we examined whether the proteasomal degradation of soluble Htt103QP affects Htt103QP IBophagy. E3 ubiquitin ligase San1 and the core Cdc48 subunits are shown to be essential for proteasomal degradation of Htt103QP [29]. In this study, we found they are also required for Htt103QP IBophagy. In addition, we identified several Cdc48 Ubx cofactors required for Htt103QP IBophagy. Strikingly, only the *ubxΔ* mutants exhibiting defective Htt103QP IBophagy showed compromised proteasomal Htt103QP degradation, thereby establishing a strong correlation between proteasomal and autophagic degradation of Htt103QP. In addition, the inhibition of proteasome activity by MG132 also blocks Htt103QP IBophagy. Together, these results support the possibility that proteasomal degradation of soluble Htt103QP is a prerequisite for IBophagy (Figure 5E).

We and others found that mHttEx1 aggregates and IBs are cleared by autophagy [8,22]. Interestingly, mHttEx1 has been shown to impair autophagy [23,24,25]. For the IBophagy protocol in this research, Htt103QP IB formation was induced by growing yeast cells in galactose medium overnight. Under this condition, we speculate the presence of both soluble Htt103QP monomers or oligomers and insoluble Htt103QP IBs. Therefore, it is likely that soluble Htt103QP inhibits IBophagy until their proteasomal degradation. This notion can explain why IBophagy does not occur for yeast cells in galactose medium. However, the addition of glucose shuts off Htt103QP production, and the degradation of soluble Htt103QP by UPS alleviates its inhibition on IBophagy (Figure 5E). The strong correlation between proteasomal degradation of Htt103QP and IBophagy revealed in this study supports this possibility.

The Cdc48 complex extracts ubiquitinated misfolded proteins from aggregates and unfolds them for proteasomal degradation [47]. In our model, the Cdc48 complex likely promotes IBophagy by facilitating the proteasomal degradation of soluble Htt103QP. Previous studies indicate that the Cdc48 complex is also involved in the autophagy of stress granules and ribosomes in budding yeast and mammalian cells [48,49]. Moreover, the Cdc48 complex promotes autophagy by facilitating autophagosome closure by extracting Atg8 with the assistance of cofactor Ubx1 [50]. Although our results support the possibility that the Cdc48 complex promotes IBophagy by facilitating the degradation of soluble Htt103QP, the Cdc48 complex may also promote IBophagy through other mechanisms.

mHttEx1 interacts with cellular membranes and disrupts lipid bilayers [51,52]. In addition, mHttEx1 IBs interact with ER membrane, with portions of ER membrane being incorporated into IBs during their formation [53,54]. As the transmembrane protein in the autophagy machinery, Atg9 scramblase collaborates with Atg2 bulk lipid transporter, which together promote autophagosome growth and formation [55,56]. Therefore, one untested possibility is that the interactions of mHttEx1 with ER membrane may compromise Atg9/Atg2-dependent vesicle transport and subsequent autophagosome formation. However, more research is needed to further define the negative role of mHttEx1 in autophagy.

UPS and autophagy are conserved protein degradation pathways, and they are critical to combat the cytotoxicity of misfolded proteins. Deficiencies in autophagy and proteasomal degradation are linked to human diseases, including neurodegenerative disorders, cancer, diabetes, and cardiovascular disease [57,58]. Therefore, enhanced clearance of misfolded proteins should be a strategy to combat the cytotoxicity of misfolded proteins. Indeed, promoting autophagic and proteasomal turnover of mHttEx1 is shown to ameliorate mHttEx1-induced toxicity in neurons [9,59]. Our results established an interesting correlation between the proteasomal degradation of mHttEx1 and autophagy. Because previous studies indicate the negative role of mHttEx1 in autophagy, it is likely that UPS-dependent degradation of soluble mHttEx1 alleviates its negative effect on autophagy. Therefore, our observation provides a new angle to understand the cytotoxicity of soluble mHttEx1 and the role of UPS in combating this toxicity.

## Figures and Tables

**Figure 1 cells-15-00068-f001:**
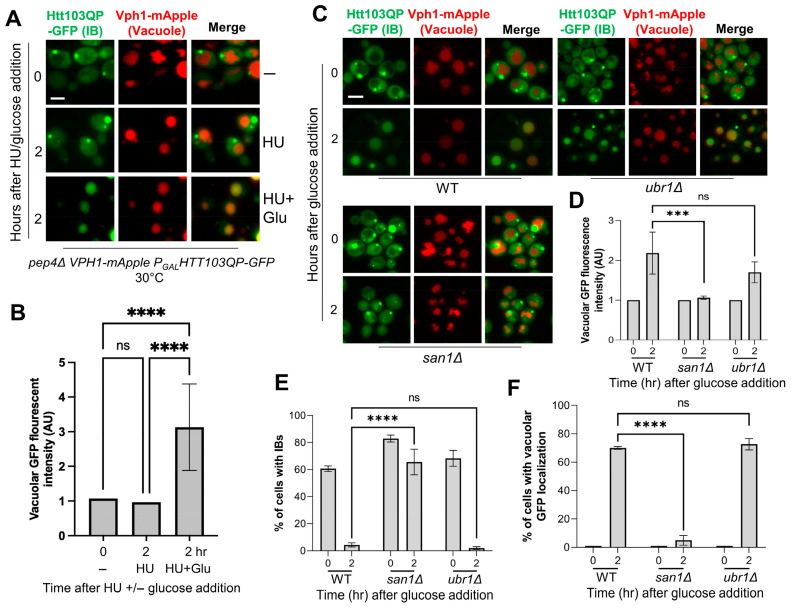
San1, the E3 ubiquitin ligase for Htt103QP, is required for Htt103QP IBophagy. (**A**) Verification of the protocol for Htt103QP IBophagy examination in yeast cells. *pep4Δ VPH1-mApple* yeast cells with integrated *P_GAL_FLAG-Htt103QP-GFP* plasmid were grown in galactose media at 30 °C for 16 h to induce Htt103QP IB formation. Cells were imaged before or after addition of 200 mM hydroxyurea (HU) alone or HU and 2% glucose for 2 h at 30 °C. Vph1-mApple marked vacuoles. Here, we show the images of Htt103QP-GFP and Vph1-mApple before (0) and after addition of HU or HU plus glucose for 2 h. Scale bar = 5 μm. (**B**) The IBophagy was quantified by measuring fluorescence intensity of GFP inside the vacuole using hybrid cell count (n > 50). Vacuolar localization of Htt103QP-GFP indicates IBophagy. Statistical significance was determined by **** *p* < 0.0001, using Sidak’s two-way ANOVA. (**C**) Htt103QP IBophagy in WT, *san1Δ,* and *ubr1Δ* cells. All yeast strains in this experiment carried *pep4Δ VPH1-mApple P_GAL_FLAG-Htt103QP-GFP*. The cells were treated to induce IBophagy as described in (**A**). Scale bar = 5 μm. (**D**) Quantification of IBophagy based on vacuolar GFP intensity. IBophagy in WT, *san1Δ*, and *ubr1Δ* mutant cells was assessed by measuring GFP intensity within the vacuole as described above. *** *p* < 0.001. (**E**) Quantification of IBophagy by determining the proportion of cells containing Htt103QP IBs. The percentage of cells with IBs was counted in WT, *san1Δ*, and *ubr1Δ* strains following IBophagy induction (n > 100). Statistical significance was evaluated using Tukey’s two-way ANOVA (**** *p* < 0.0001). (**F**) IBophagy quantification by counting cells exhibiting vacuolar GFP. The percentage of cells with vacuolar GFP signal was quantified for each strain (n > 100), and statistical analysis was performed as described above.

**Figure 2 cells-15-00068-f002:**
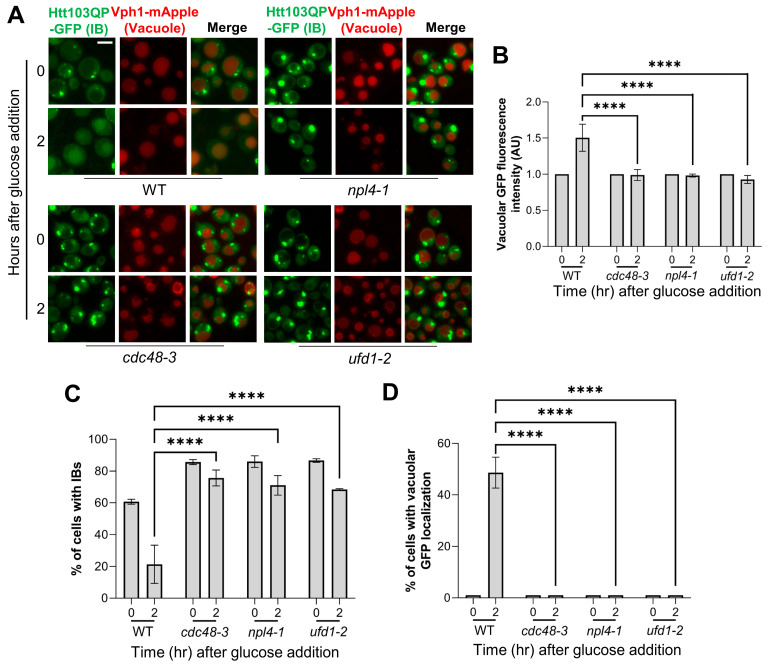
Htt103QP IBophagy depends on the Cdc48 complex. (**A**) Htt103QP IBophagy in WT and Cdc48 complex mutants. Strains with indicated genotypes were cultured overnight in galactose-containing media at 25 °C to induce Htt103QP IB formation. Cells were then shifted to 37 °C for 1 h to inactivate the Cdc48 complex, followed by the addition of glucose and HU. Representative images of Htt103QP-GFP and the vacuole (Vph1-mApple) are shown before (0 h) and after IBophagy induction (2 h). Scale bar, 5 μm. (**B**) Quantification of IBophagy based on vacuolar GFP intensity. GFP fluorescence intensity within the vacuole was measured using hybrid cell count analysis (n > 50). Statistical significance was determined by Tukey’s two-way ANOVA (**** *p* < 0.0001). (**C**) Quantification of IBophagy by determining the proportion of cells containing Htt103QP IBs (n >100). Statistical significance was assessed using Tukey’s two-way ANOVA (**** *p* < 0.0001). (**D**) Quantification of IBophagy by calculating the percentage of cells exhibiting vacuolar GFP signal (n > 100). Statistical analysis was performed as described above.

**Figure 3 cells-15-00068-f003:**
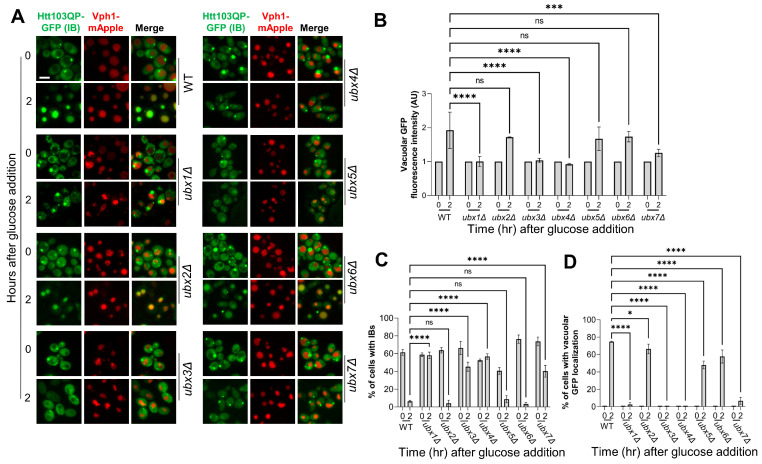
Specific Ubx cofactors are required for Htt103QP IBophagy. (**A**) Htt103QP IBophagy in WT and *ubxΔ* mutants. WT and mutant strains in *pep4Δ VPH1-mApple P_GAL_FLAG-Htt103QP-GFP* background were cultured in galactose-containing medium at 30 °C for 16 h to induce Htt103QP IB formation. Glucose and HU were added to trigger IBophagy and yeast cells were imaged before (0 h) and after HU/glucose addition for 2 h. Representative images of Htt103QP–GFP and Vph1-mApple are shown. Scale bar, 5 µm. (**B**) Quantification of IBophagy by measuring vacuolar GFP intensity in 50 cells using hybrid cell count analysis. Statistical significance was determined by Tukey’s two-way ANOVA (*** *p* < 0.001, **** *p* < 0.0001). (**C**) Quantification of IBophagy by determining the percentage of cells containing Htt103QP IBs (n > 100). Statistical significance was assessed using Tukey’s two-way ANOVA (**** *p* < 0.0001). (**D**) Quantification of IBophagy by calculating the percentage of cells with vacuolar GFP signal (n > 100). Statistical analysis was performed as described above. * *p* < 0.05, **** *p* < 0.0001.

**Figure 4 cells-15-00068-f004:**
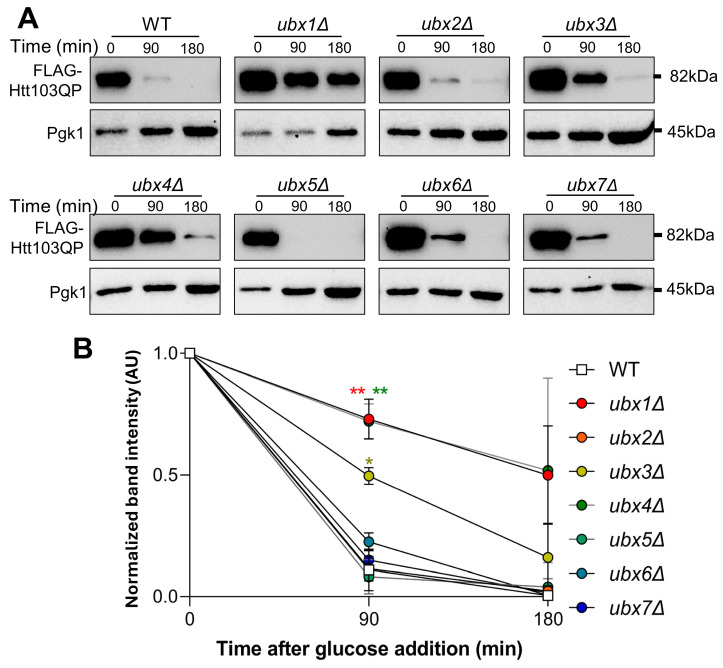
Certain Ubx cofactors of Cdc48 are required for efficient proteasomal degradation of Htt103QP. (**A**) Proteasomal degradation of Htt103QP in WT and *ubxΔ* mutants. WT and mutant cells with *pep4Δ VPH1-mApple P_GAL_FLAG-Htt103QP-GFP* were grown in raffinose medium overnight at 30 °C to log phase. Htt103QP production was induced by adding galactose for 1 h before shut-off with glucose. Samples were taken before (0) and after glucose addition (90 and 180 min). Western blotting was performed with anti-FLAG antibody to monitor Htt103QP protein levels. Pgk1, loading control. (**B**) Proteasomal degradation of Htt103QP was quantified by measuring the ratio of Htt103QP band intensity over Pgk1 from three repeats using ImageJ (National Institute of Health, 1.54f). The line graph shows the relative intensity of Htt103QP protein bands from three repeats of Western blotting. Statistical significance was determined by * *p* < 0.05, ** *p* < 0.01, using Dunnett’s two-way ANOVA.

**Figure 5 cells-15-00068-f005:**
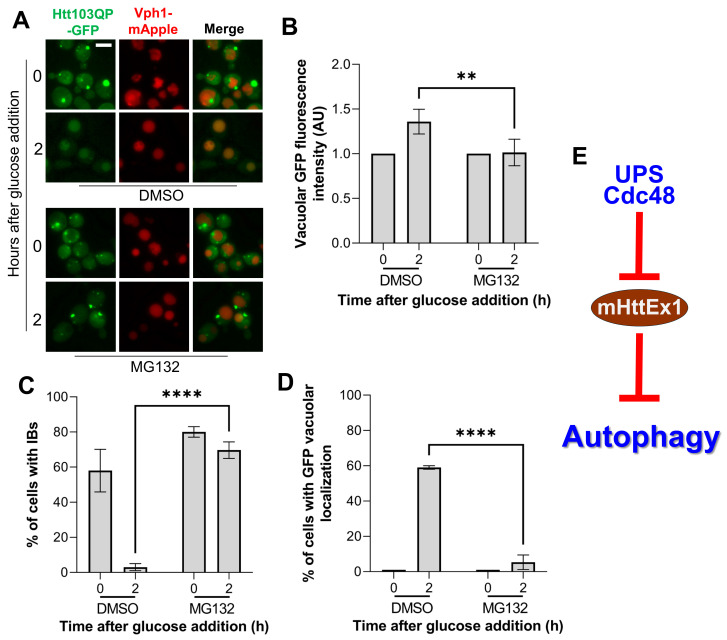
Functional proteasomes are required for Htt103QP IBophagy. (**A**) *pep4Δ VPH1-mApple P_GAL_FLAG-Htt103QP-GFP* cells were grown to log phase overnight at 30 °C in galactose media to induce Htt103QP IB formation. After treatment of yeast cells with L-proline and SDS for 3 h to increase permeability, proteasome inhibitor MG132 was added to 75 μM for 30 min. DMSO was used as a control. Then, glucose was added to induce IBophagy. Here, we show images of Htt103QP-GFP and the vacuole (Vph1-mApple) before and after IBophagy induction for 2 h. Scale bar = 5 μm. (**B**) Quantification of IBophagy based on vacuolar GFP intensity. IBophagy in DMSO- or MG132-treated cells was assessed by measuring GFP intensity within the vacuole in 50 cells using hybrid cell count analysis. Statistical significance was evaluated using Sidak’s two-way ANOVA (** *p* < 0.01). (**C**) Quantification of IBophagy by determining the percentage of cells containing IBs (n > 100). Statistical significance was evaluated using Sidak’s two-way ANOVA (**** *p* < 0.0001). (**D**) Quantification of IBophagy by calculating the percentage of cells exhibiting vacuolar GFP signal (n > 100). Statistical analysis was performed as described above. (**E**) Proposed working model. Soluble mHttEx1 suppresses autophagy, whereas its proteasomal degradation by UPS/Cdc48 pathway relieves this inhibition.

## Data Availability

The original contributions presented in this study are included in the article/Appendix A. Further inquiries can be directed to the corresponding author.

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
