# Peer review of "Proteasomal Degradation of Mutant Huntingtin Exon1 Regulates Autophagy"

_cells, 2025, doi:10.3390/cells15010068_

Round 1
Reviewer 1 Report
Comments and Suggestions for Authors
This is a very well-written manuscript describing some very interesting discoveries regarding the potential cooperation between proteasome and vacuole in removing mHtt. The data are of high quality and significant. I only have some very minor comments for the authors to consider.
- When the authors describe their work on the involvement of proteasome in degrading mHtt, it would be helpful to reiterate that pep4 was deleted to prevent vacuolar degradation and thus any degradation observed can be attributed to proteasome.
- Normally, yeast cells do not respond to MG132, unless certain genes being deleted or special treatment being performed. The authors should briefly mention how they have made yeast cells responding to MG132 in their study. Ideally, for Fig 5, a western blot with anti-ubiquitin demonstrating MG132 actually works would be reassuring.
Reviewer 2 Report
Comments and Suggestions for Authors
The manuscript investigated the correlation between ubiquitin-proteasome system (UPS) and autophagic clearance of inclusion bodies (IBophagy) in budding yeast. The authors found that the proteasomal degradation of mutant huntingtin (mHtt) is associated with the mHtt IBophagy and identified that San1, Cdc48 complex and certain Ubx proteins are involved in this process. This is a preliminary study but with substantial data supporting the conclusion temporally.
- Based on the results from yeast model, the authors can draw a conclusion that the UPS is contributable to the autophagic clearance of mHtt IBs (IBophagy). However, a schematic illustration (or graphical abstract) is needed to clearly delineate the mechanism underlying mHtt IB clearance and cytotoxicity.
- Figure 5 is not indicated in the Result section of “3.4 Proteasome function is required for Htt103QP IBophagy”.
- Figures 1C, 2C, 5C. To present the data clearly, it is suggestive to split the panel C into two subfigures, labeling “C” and “D” individually and describing them separately in the context.
- The term “huntingtin” is for a protein that leads to Huntington’s disease; it may not use initial capitalization, but it is commonly abbreviated to “Htt”.
Reviewer 3 Report
Comments and Suggestions for Authors
Rec Folger et al., 2025
Studying degradation of N-terminal fragment of mutant huntingtin as a model of Huntington diseases in yeast authors show that San1 ubiquitin ligase, components of Cdc48 complex and Ubx1,3 and 4 cofactors are required for autophagy of inclusion bodies, which are aggregates formed by this protein. Interestingly, active proteasome is also required for this process. Authors propose that proteasomal degradation of soluble monomeric form of the protein studied is required for autophagic clearance of inclusion bodies. Manuscript requires several corrections.
General comments
From the Abstract readers can understand that mHtt is full length protein. Later mHtt abbreviation is used for N-terminal fragment of mHtt. Make it clear what is what.
Genes are expressed (HTT) but proteins are synthesized or produced (Htt). Genes encode proteins. Plasmids are bearing genes, not proteins. All other statements are laboratory jargon.
Consider to write full name, huntingtin, starting with small letter, like other proteins (actin, tubulin etc.).
Some strains presented in Table S1 were constructed in this work. This construction is not described in chapter 2.1. of Materials and Methods. Selective medium which is needed to construct the strains is not described. Complete your text.
Hydroxyurea, which was used in experimental procedure to stop the cell cycle before adding galactose to shut down GAL1 promoter, is causing replication stress and this induces autophagy (Cecconi F. (2020) Cell Death and Differentiation 27 829-830; Bartek et al. (2020) Cell Death and Differentiation 27; 1134-1153). The control with hydroxyurea but without galactose is missing in presented experiments. To stop the cell cycle better use alpha-factor which does not cause autophagy induction by itself.
Moreover, the Saccharomyces Genome Database gives the information that VPH1 /YOR270Cn encodes ”subunit a of the vacuolar-ATPase V0 domain; …; relative distribution to the vacuolar membrane decreases upon DNA replication stress;” which indicates that maybe Vph1 is not the best marker in these experiments.
Specific comments
P1, L20 in selective autophagy
Abstract
P1, L24 proteins accumulation
P1, L26 results in the production of mutant Huntingtin (mHtt) containing polyglutamine tracks that is prone to cytotoxic aggregation.
P1, L30 ubiquitinated protein is first recognized (one by one)
P1, L36 regulation and cytotoxicity of mHtt.
Introduction
P2, L;48 include the degradation of toxic protein by ubiquitin-proteasome system and macroautophagy.
P2, L50 to the accumulation of misfolded proteins.
P2, L53 in exon 1 develop Huntington’s disease.
P2, L54-56 CAG repeats in HTT result in production of extended Htt proteins that are prone to aggregation. Explain what is the source of the N-terminal Htt fragments, proteolytic processing or what? Here new abbreviation appears: N-terminal Htt fragments containing the polyQ repeats are (mHtt ). Make it clear, mHtt is full length or N-terminal fragment of Htt?
P2, L58 mHtt production
P2, L65 which fuse with the vacuole and deliver cargo for vacuolar degradation. Autophagosomes are not degraded in the vacuole. Autophagosomes are fused with the vacuole and their content is degraded.
P2, L83 SUMOylation system
P2, L83 Cdc48 complex is not a “trimeric complex”. Describe better what is Cdc48 complex. Give reference.
“Cdc48 adenosine triphosphatase (ATPase), which forms a hexameric assembly that pulls polypeptides through its central pore.” (Twomey et al., Science 365, 6452, 2019, for example)
P2, L86 unfolds them and transfers to proteasome
P3, L94 UPS components
Materials and Methods
P2, L112 The integrating plasmid containing a gene, PGAL1Flag-Htt103QP-GFP under a galactose-inducible promoter, which encodes FLAG- and GFP-tagged Htt103QP fragment was from the Lindquist lab. Plasmid does not contain a protein. Plasmid is bearing a gene which encodes a protein.
P2, L111-115 add how strains were constructed in this work. Make clear that gene fusion was integrated into the genome.
P2, L120 to suppress GAL1 promoter and Htt103QP production
P2, L122 hydroxyurea is causing replication stress and this stress induces autophagy.
Results
P4, L152 remove ” for Htt1003QP”, not needed, shorter title is better.
P4, L153 after overnight induction of respective gene expression and production of respective protein. Galactose does not induce the protein. Be precise.
P4, L159-161 which contains/bears the first exon of HTT gene with 103 CAG expansion in fusion with FLAG and GFP gene encoding FLAG-Htt103QP-GFP, hereafter Htt103QP. To analyze autophagy, yeast cells also produce Vph1-mApple as a vacuolar marker, and are lacking the Pep4 protease (pep4Δ) to avoid protein degradation inside the vacuole. (exon is part of the gene, not protein)
P4, L165 Htt103QP production
P4, L173 no significant increase was observed in san1 cells.
Figure 1.
Panels are too small
There were no cells with both, IB and vacuolar signal?
P5, L182 yeast strains with gene fusion. Make clear that this gene fusion was integrated.
P5, L184 to repress HTT-103QP. Gene is repressed
P5, L198 into pep4Δ cells producing Htt103QP and Vph1-mApple.
P5, L200 IB formation and shifted to 37oC
P5, L206 cells with defective Cdc48 complex
P5, L207 remove “also” or “both”
Figure 2
Panels too small. Not readable 1:1.
P6, L 230, specific Ubx cofactors, Ubx1, Ubx3 and Ubx4,
Figure 3
Panels too small.
P7, L236 examined after 2h.
P7, L245 How the authors know that the Western blot in Figure 4 shows only proteasomal degradation and not autophagic degradation? Why no experiment with MG132 in the same condition is shown?
Figure 4
Add description in B on the left, what signal?
P8, L275 Htt103QP or some other protein
Discussion
P9, L290 Htt103QP production
P10, L325 too old references for Atg9 function. More is known, for example:
Jiyoung Choi, Haeun Jang, Zhao Xuan & Daehun Park (2024) Emerging roles of ATG9/ATG9A in autophagy: implications for cell and neurobiology, Autophagy, 20:11, 2373-2387, DOI: 10.1080/15548627.2024.2384349.
Kazuaki Matoba et al., Atg9 is a lipid scramblase that mediates autophagosomal membrane expansion Nat Struct Mol Biol. 2020 Dec;27(12):1185-1193. doi: 10.1038/s41594-020-00518-w. Epub 2020 Oct 26.
P10, L328 Atg9 is a scramblase and collaborates with Atg2 bulk lipid transporter which together promote autophagosome growth/formation
P10, L331-337 Give conclusions and perspectives and convince the reader better what is the significance of new findings.
Round 2
Reviewer 3 Report
Comments and Suggestions for Authors
The manuscript was greatly improved and is now suitable for publication. However, one satatement still needs correction.
Materials and methods
P3, L189 after linearization and integrants (or cells with integration) were selected using URA3 marker and uracil dorpout plates.